# A Repurposed Drug Selection Pipeline to Identify CNS-Penetrant Drug Candidates for Glioblastoma

**DOI:** 10.3390/ph17121687

**Published:** 2024-12-14

**Authors:** Ioannis Ntafoulis, Stijn L. W. Koolen, Olaf van Tellingen, Chelsea W. J. den Hollander, Hendrika Sabel-Goedknegt, Stephanie Dijkhuizen, Joost Haeck, Thom G. A. Reuvers, Peter de Bruijn, Thierry P. P. van den Bosch, Vera van Dis, Zhenyu Gao, Clemens M. F. Dirven, Sieger Leenstra, Martine L. M. Lamfers

**Affiliations:** 1Department of Neurosurgery, Brain Tumor Center, Erasmus MC Cancer Institute, Erasmus University Medical Center, 3015 GD Rotterdam, The Netherlandsc.dirven@erasmusmc.nl (C.M.F.D.); s.leenstra@erasmusmc.nl (S.L.); 2Department of Medical Oncology, Erasmus MC Cancer Institute, Erasmus University Medical Center, 3015 GD Rotterdam, The Netherlands; 3Department of Hospital Pharmacy, Erasmus University Medical Center, 3015 GD Rotterdam, The Netherlands; 4Division of Pharmacology, The Netherlands Cancer Institute, 1066 CX Amsterdam, The Netherlands; 5Department of Neuroscience, Erasmus Medical Center, 3015 GD Rotterdam, The Netherlandsz.gao@erasmusmc.nl (Z.G.); 6Department of Radiology & Nuclear Medicine, Erasmus Medical Center, 3015 GD Rotterdam, The Netherlands; 7Department of Molecular Genetics, Erasmus Medical Center, 3015 GD Rotterdam, The Netherlands; 8Department of Pathology, Erasmus Medical Center, 3015 GD Rotterdam, The Netherlands; t.vandenbosch@erasmusmc.nl (T.P.P.v.d.B.);

**Keywords:** glioblastoma, BBB, ABC transporters, CNS-MPO, pharmacokinetics, PDX models

## Abstract

Background: Glioblastoma is an aggressive and incurable type of brain cancer. Little progress has been made in the development of effective new therapies in the past decades. The blood–brain barrier (BBB) and drug efflux pumps, which together hamper drug delivery to these tumors, play a pivotal role in the gap between promising preclinical findings and failure in clinical trials. Therefore, selecting drugs that can reach the tumor region in pharmacologically effective concentrations is of major importance. Methods: In the current study, we utilized a drug selection platform to identify candidate drugs by combining in vitro oncological drug screening data and pharmacokinetic (PK) profiles for central nervous system (CNS) penetration using the multiparameter optimization (MPO) score. Furthermore, we developed intracranial patient-derived xenograft (PDX) models that recapitulated the in situ characteristics of glioblastoma and characterized them in terms of vascular integrity, BBB permeability and expression of ATP-binding cassette (ABC) transporters. Omacetaxine mepesuccinate (OMA) was selected as a proof-of-concept drug candidate to validate our drug selection pipeline. Results: We assessed OMA’s PK profile in three different orthotopic mouse PDX models and found that OMA reaches the brain tumor tissue at concentrations ranging from 2- to 11-fold higher than in vitro IC_50_ values on patient-derived glioblastoma cell cultures. Conclusions: This study demonstrates that OMA, a drug selected for its in vitro anti-glioma activity and CNS- MPO score, achieves brain tumor tissue concentrations exceeding its in vitro IC_50_ values in patient-derived glioblastoma cell cultures, as shown in three orthotopic mouse PDX models. We emphasize the importance of such approaches at the preclinical level, highlighting both their significance and limitations in identifying compounds with potential clinical implementation in glioblastoma.

## 1. Introduction

Glioblastoma, one of the most lethal forms of cancer, continues to pose a significant challenge in the medical field in spite of intense efforts to introduce new treatments [1,2]. This is largely attributed to the intra- and intertumoral heterogeneity that contributes to the tumor’s resistance to therapeutic treatments [3,4]. Additionally, the highly infiltrative nature of glioma cells allows them to invade healthy brain parenchyma and form malignant “foci” in regions protected by an intact BBB, thereby further complicating treatment strategies [5]. Consequently, the BBB poses a major obstacle in the implementation of promising drug candidates to the clinical setting, presenting a significant impediment to therapeutic advancements for this tumor type [6].

In glioblastoma, the BBB, also known as the blood–tumor barrier (BTB), is compromised, as manifested by contrast enhancement in T1W post-gadolinium MRI scans. However, the degree of disruption differs per location within the tumor, with areas closer to the tumor rim often being less BBB permeable compared with more central areas of the tumor [7]. Additionally, drug efflux transporters play an important role in the poor response of glioblastoma to pharmacotherapies. In particular, ABC transporters such as P-glycoprotein (P-gp; ABCB1) and breast cancer resistance protein (BCRP; ABCG2) restrict the brain accumulation of many compounds [8,9]. These transporters are expressed at the BBB and are involved in the active efflux of drugs from the brain [8]. De Gooijer et al. reported that ABC transporters play a major role in the restriction of drug accumulation in brain tumors even when the BBB is disrupted [7,10]. Additionally, factors such as plasma protein binding may influence the percentage of the free drug delivered into the brain. These findings may explain the failure in systemically administered drugs to treat brain tumors and highlight the importance of identifying novel candidates not only with optimal PK profiles for CNS penetration but also with limited affinity for efflux pumps [11,12,13].

To improve the discovery and development of compounds for CNS penetration, various in silico approaches such as CNS-MPO, SwissADME tools and physiology-based PK models have been developed to score compounds based on their physicochemical properties and to predict promising CNS drug candidates [14,15,16,17]. In recent years, drug repurposing, a drug discovery approach that re-evaluates available anti-cancer agents for anti-glioma activity and favorable properties for BBB penetration, is gaining attention [1]. We previously reported on a set of compounds that are effective against isocitrate dehydrogenase 1 (IDH1) mutant glioma, including omacetaxine mepesuccinate ((OMA) formerly known as homoharringtonine). This anti-cancer agent was granted FDA approval in 2012 for the treatment of hematological malignancies (CMLs) in patients with relapse after the use of one or more kinase inhibitors [18,19]. Recently, OMA was found to exhibit a strong anti-cancer effect also on solid tumors, such as hepatocellular carcinoma (HCC) [20,21]. OMA mainly acts through the inhibition of protein synthesis by preventing aminoacyl tRNA from binding to the ribosomes during mRNA translation [22]. In both hematological and solid tumors, OMA was found to affect multiple cellular processes, including the induction of apoptosis and cell cycle arrest, and the inhibition of multiple oncoproteins such as cyclin-D1, β-catenin, XIAP, MET and c-Myc [20,22,23]. In glioblastoma, various in vitro drug screening studies have identified OMA as a potent anti-glioma agent in both isocitrate dehydrogenase (IDH1) wildtype and mutant tumors [24,25,26,27].

In this study, we employed the FDA-approved Oncology Drug Set II library consisting of 107 chemotherapeutic drugs and ranked them based on their in vitro activity and physicochemical properties for CNS penetrations [24]. The main goal of this study was to validate the drug delivery of a candidate drug selected based on these criteria. OMA ranked among the top candidates and, considering available promising preclinical data, we chose to investigate its PK profile in three different orthotopic PDX models of glioblastoma. These models were chosen for their representation of different growth rates, the distinct glioblastoma subtypes and their ability to mimic the invasive characteristics of glioma cells. Prior to assessing OMA’s PK profile, we examined the BBB integrity and ABC transporter expression in each model.

Our study demonstrates the importance of using PDX models with varying BBB characteristics to evaluate the PK profiles of drug candidates for CNS penetration and to determine the range of drug concentration within brain tumor tissue. In addition, we highlight the importance of ABC transporters in drug delivery across the BBB. By evaluating the PK profile of omacetaxine mepesuccinate using our model system, we demonstrated that our drug selection approach effectively identifies candidates with brain tumor penetrant properties.

## 2. Results

### 2.1. Subsection Selection of Candidate Drug

As previously described by Verheul et al. [24], we evaluated the physicochemical properties of drugs with anti-glioma activity selected from 107 FDA approved oncological agents using a CNS-MPO algorithm [16]. The cut-off value of the MPO score for this selection was ≥4. Additionally, we conducted the literature research and gathered information regarding the affinity of candidate drugs for the ABCB1 and ABCG2 transporters and the percentage of plasma protein (albumin) binding and reported Cmax plasma values in patients [28,29,30] in DrugBank. We focused on drugs that are negative substrates for the ABC transporters and have an albumin binding below 50%. OMA was ranked among the top candidates based on this approach and was selected for further investigation (Appendix A).

### 2.2. In Vitro Activity of Omacetaxine

The sensitivity to OMA of three glioblastoma patient-derived cell cultures, GS607, GS832 and GBM8, was assessed. Comparing the in vitro IC_50_ values of OMA with the reported Cmax plasma value (43 nM) in humans, we found that GS832 and GBM8 cell lines are sensitive to OMA with IC_50_ values of 4.9 nM and 7.2 nM, respectively, while GS607 is the most resistant among these cell cultures with an IC_50_ of 22.9 nM (Appendix A).

### 2.3. Omacetaxine Crosses the BBB and Accumulates in the Mouse Brain Tumor Tissue

OMA is a natural cephalotaxine alkaloid and primarily hydrolyzed to 4′-DMHHT through plasma esterases with hepatic microsomal oxidative and/or esterase-mediated metabolism [31]. In humans, the C_max_ plasma of OMA can be reached in 1 h when the drug is administered s.c., whereas the half-life is 7 h [32]. The elevated number of esterases in mouse plasma can affect the metabolic profile of OMA. Therefore, using LC-MS/MS, we determined the concentration of OMA in both plasma and brain and liver tissue of mice bearing GS832 tumors after a single s.c. dose of 1.5 mg/kg OMA (Figure 1). We observed the peak concentration of OMA in plasma at 15 min (mean 396 ± 59 ng/mL), which subsequently decreased rapidly to 166 ± 29 ng/mL at 60 min and 24 ± 3 ng/mL at 240 min post-drug administration, which yielded a half-life in mice of 50 min (Table 1).

Similar to the plasma concentrations, we observed a rapid decrease in the mean concentration of OMA in the liver between 60 min (680 ± 156 ng/g) and 240 min (190 ± 48 ng/g) (Figure 1B). In the brain, the total mean OMA concentration (18 ± 6 ng/g) was approximately 40-fold lower than in the liver and decreased more slowly in time (Figure 1C); hence, the mean brain-to-plasma (B/P) ratio increased from 0.10 (± 0.05) at 60 min, to 0.50 (± 0.33) at 240 min (Table 2). Concentrations in the tumor-bearing hemisphere were not significantly different from the contralateral hemisphere. (Figure 1D).

Next, we assessed OMA’s PK profile in two additional glioblastoma PDX models, GS607 and GBM8, ands in naïve non-tumor-bearing mice. Based on the previous findings, we measured the drug concentration in the blood at 30 and 60 min (Figure 1E,I) and tissue samples at 60 min after a single dose of 1.5 mg/kg OMA. Mean plasma concentrations at 60 min were 88 ± 9 ng/mL and 391 ± 58 ng/mL for GS607-PDX and GBM8, respectively. The mean liver drug concentration was 1653 ± 163 ng/g and 998 ± 254 ng/g, for GS607-PDX and GBM8, respectively, while 715.9 ± 4.6 ng/g for naïve mice (Figure 1F,J). In the brain tumor tissue, OMA reached higher mean concentrations in tumor-bearing hemispheres of GBM8 (45 ± 20 ng/g) than in GS607-PDX (21 ± 5 ng/g) or GS832-PDX (18 ± 6 ng/g) models (Figure 1G,K, Table 2). As expected, the lowest drug concentration was measured in the naïve mice (16.4 ± 2.9) (Table 2). The measured brain tumor concentrations corresponded with 82.5 nM, 37.9 nM and 33.0 nM for GBM8, GS607 and GS832, respectively (Appendix A). Interestingly, despite the established leakiness of the BTB, we did not detect a significant difference in OMA concentrations in the tumor-bearing versus contralateral hemispheres in any of the tested models (Figure 1C,G,K). The B/P ratio of OMA at 60 min post-injection did not vary considerably among the three PDX models; 0.10 for GS832, 0.23 for GS607 and 0.11 for GBM8 (Figure 1D,H,L, Table 2). Overall, these mouse PK data indicate that OMA has brain tumor-penetrating properties reaching higher tumor concentrations than the IC_50_ values found in vitro in GSCs (Appendix A).

### 2.4. Characterization of the BBB Permeability in Orthotopic Mouse PDX Models

Drug delivery into brain tumor tissue in pharmacologically effective concentrations is a challenging process that is dependent on the vascular integrity of the brain capillaries and the leakiness of the BBB. To gain insight into these properties in the three different PDX model systems, we evaluated both the histology and the extent of the BBB disruption by T1-weighted (TW1) post-gadolinium (Gd) MRI scans (Figure 2 and Appendix A). Tumor presence in the mouse brains was confirmed histologically by hematoxylin/eosin (H/E) staining. The GS607-PDX, derived from a mesenchymal subtype of glioblastoma, presents histologically with diffusely infiltrative glial cells, large atypical nuclei and mitosis and signs of microvascular proliferation. Similarly, GS832-PDX, derived from a proneural subtype of glioblastoma, presents with signs of gliosis with hyperchromatic nuclei and mitosis. Lastly, GBM8, the fastest growing and most aggressive of the three models, was previously characterized as a neural subtype with a subfraction of mesenchymal cells [33]. It is characterized by diffuse infiltrative glioma cells and extensive mitosis (Appendix A).

In terms of BBB disruption, GS607-PDX revealed a ring-enhancing lesion on the TW1 + Gd, a typical characteristic of glioblastoma and indicative of increased interstitial pressure in the tumor. In contrast, TW2/TW1 scans of GBM8 and GS832 displayed clear tumor lesions, including a clear midline shift in GBM8 (Appendix A). However, following a gadolinium injection, minimal or no tumor enhancement was observed (Figure 2). Taken together, GS607 showed a degree of BBB leakiness, while this was very limited for GBM8 and GS832.

### 2.5. Characterization of Vascular Integrity in Orthotopic Mouse PDX Models

The vascular integrity of brain capillaries in each PDX model was assessed by immunohistochemical (IHC) staining for zonula occludens-1 (ZO-1), a tight junction marker that is used to assess vascular integrity, and glucose transporter-1 (GLUT-1), an endothelial marker for the brain capillaries. Additionally, we stained for Nestin to detect the presence of stem/progenitor cells in the brain tumor tissue (Figure 3). In GS607-PDX, the expression of GLUT-1 indicated that the tumor was vascularized; however, the observed brain capillaries displayed a highly irregular morphology. Interestingly, we did not observe co-localization of ZO-1 with GLUT-1, but did observe co-localization of ZO-1 with Nestin, suggesting junction features between the stem/progenitor cells within the tumor. The expression of GLUT-1 in the absence of co-expression with ZO-1 suggested that the capillaries lacked tight junctions and were permeable (Figure 3). The latter was further supported by the TW1-MRI scans indicating a leaky BBB (Figure 2).

The GS832-PDX model formed more compact Nestin-positive tumors, the low expression of GLUT-1, indicating the relatively limited presence of blood vessels within this tumor, while the co-localization of ZO-1 expression suggested that their vascular integrity was preserved (Figure 3), coinciding with a lack of gadolinium leakage into the tumor (Figure 2). Finally, in GBM8-PDX, the highest expression of GLUT-1 among the three PDX models was found, revealing large and regularly shaped intratumoral vasculature. Interestingly, the lack of co-localized expression with ZO-1 on the capillaries in this tumor suggested loss of tight junctions and partial loss of BBB integrity (Figure 3); however, this was not supported by the TW1 + Gd (Figure 2). Taken together, these findings indicate variability in the BBB permeability among the tested PDX models in this study, which cannot be solely explained by the degree of vascularization and barrier integrity (Appendix A).

### 2.6. Both P-gp and BCRP Transporters Are Expressed in All Three PDX Models and Omacetaxine Is a Weak Substrate for P-gp

The ATP-binding cassette transporters play a key role in the accumulation of a drug, and high expression of these transporters is linked to the development of multidrug resistance. Therefore, we assessed the expression of the transporter proteins in the tumors of the three PDX models by immunohistochemistry for P-gp (ABCB1) and BCRP (ABCG2). Notably, all three glioblastoma PDX models expressed both P-gp and BCRP transporters on the brain capillaries but not on the glioma cells (Figure 4A).

Furthermore, to investigate the impact of efflux pumps on OMA’s PK profile and accumulation in the brain tumor tissue, we assessed the affinity of this drug for the human P-gp (ABCB1) and BCRP (ABCG2) transporters and the murine variants (Abcb1a and Abcg2) using a Transwell filter transporter assay. As shown in Figure 4B, OMA’s concentration in the apical compartment of LLC-ABCB1 and LLC-Abcb1a cells increased in time at the expense of the concentration in the basolateral compartment; however, the difference between these two compartments was modest. The parental LLC-PK1 line translocation of OMA from basolateral to apical was even smaller. This line is known to express some endogenous porcine ABCB1, which for stronger substrates can result in relatively high translocation. Thus, OMA appears to be a weak substrate of human and murine P-gp (ABCB1/Abcb1).

In the MDCK cell lines, zosuquidar was used to inhibit any endogenous activity of canine P-gp. The results in this line suggest that there is no meaningful translocation of the drug between the apical and basolateral compartments of MDCK-ABCG2 or MDCK-Abcg2 cells, indicating that OMA is not a substrate of BCRP (ABCG2/Abcg2) (Figure 4B).

## 3. Discussion

Identifying drugs at a preclinical level with optimal properties for CNS penetration remains a major challenge. Intracranial glioblastoma PDX models were found to recapitulate the BTB of glioblastoma and therefore have been extensively used to assess the PK/PD profile of various therapeutic agents [34]. The increased leakiness of the BTB often allows chemotherapeutic agents to reach brain tumors, although recent studies have proven that drug accumulation is not solely linked to the level of BBB disruption [35,36]. Factors such as brain vascular integrity, expression of efflux transporters and the percentage of unbound drug (for systemically administered compounds) can significantly affect the delivery of a drug into the brain tumor tissue [37].

In brain tumors, the downregulation of the expression of tight junctions, proteins responsible for maintaining the integrity of the brain vasculature, is linked to an increased permeability of the BBB [38]. The level of BBB permeability is strongly associated with the location of the vasculature within the tumor. The core tumor regions are more permeable than the peri-tumoral regions, with abnormal vascular structures showing loss of tight junctions and decreased astrocytic end-feet connections and pericyte function (1). Moreover, the expression of efflux transporters on the endothelial cells of brain (tumor) vasculature and on the glioma cells has been found to significantly limit the accumulation of drugs with high affinity for these transporters [39]. To date, the most widely studied efflux pumps are P-gp and BCRP, both members of the ABC transporter family. These transporters have been found to play a crucial role in the development of multidrug resistance and in the clinical failure of many compounds tested against glioblastoma [40]. Another factor often neglected when it comes to the systemic administration of CNS-targeted compounds is the percentage of plasma protein binding. In general, compounds need to obtain certain physicochemical properties that will allow them to reach the CNS. Such properties include molecular weight (MW), topological polar surface area (TPSA), number of hydrogen binding donors (HBD), constant of dissociation (pKa), constant of distribution (logD) and lipophilicity [16]. The latter allows molecules to cross the cellular membranes of endothelial cells through passive diffusion. However, systemically administered drugs with a very high lipophilic profile will lead to an increased plasma protein binding (mainly albumin) and subsequently to a decreased percentage of the unbound (active) drug in the circulating system. Another important physicochemical property that can influence the drug distribution is the number of HBD. Compounds with a low number of HBD (e.g., ≤3) have sufficient polarity for solubility and bioavailability.

Taking these factors into consideration and building upon previously obtained drug sensitivity data [24], we selected OMA for a PK profiling study in three different intracranial glioblastoma PDX mouse models. These PDX models were characterized in terms of brain vascular integrity, BBB permeability and expression of ABC transporters to allow us to understand how these parameters affect the drug accumulation in the brain tumor tissue. In order to assess whether the accumulated concentration of OMA in the brain tumor tissue would reach therapeutic concentrations, we also determined the OMA sensitivity of the three glioblastoma cell lines used in these in vivo experiments.

OMA reached the highest concentration in plasma at 15 min post-administration, followed by a rapid 17-fold reduction in the concentration by 4 h. The half-life of the drug in mice is therefore much shorter than in humans (50 min versus 7 h, respectively). This underscores the impact of a higher metabolic rate in mice and highlights one of the limitations of the mouse models.

The highest brain tumor tissue drug concentrations were measured in the GBM8 model and the lowest in the naïve non-tumor-bearing mice (Table 2). The variation in OMA’s systemic distribution between orthotopic PDX models and naïve mice may be attributed to glioblastoma’s effects on systemic blood flow and organ function, particularly if the tumor elevates intracranial pressure or triggers stress responses. These factors could influence circulation and subsequently alter the distribution and metabolism of omacetaxine in other tissues. Additionally, the inter-model variability in the accumulation of OMA can be explained by the unique characteristics of each model in terms of BBB permeability. The GS607 model forms less infiltrative tumors with a general loss of the tight junctions leading to a leaky BBB, as visualized in the T1W + Gd images. Similarly, GBM8 is a highly vascularized tumor with regularly shaped vasculature. Despite the absence of ZO-1 staining, indicating the potential dysregulation of vascular integrity, this tumor did not reveal leakiness on the T1W + Gd images. Of note, ZO-1 is a key component of the tight junction architecture within the BBB, but it is only one of several tight junction proteins; others, such as VE-cadherin, occludin and claudin-5, also play critical roles in maintaining barrier integrity. Finally, GS832 exhibits the least permeable BBB among these models, with distinct tight junctions on its vascular structures. The lack of BBB leakiness in this model, as predicted from the MRI imaging, corresponds to the lower concentrations of OMA measured in the mouse brain tumor tissues. Despite the variable OMA levels found in the three PDX models, tumor concentrations at 60 min ranged from 1.7- to 11-fold higher than OMA IC_50_ values found in vitro, suggesting that therapeutically active concentrations can be reached, albeit potentially only for a short period of time (Appendix A).

OMA was found to be a weak substrate for P-gp. Additionally, all three PDX models exhibited similar expressions of P-gp exclusively on the vascular structures, with no expression on the tumor cells. Nonetheless, the transportation of compounds, even those with a low affinity for P-gp, can still be influenced when these compounds are water-soluble. Low lipid permeability impedes inward-directed passive diffusion, making it easier for P-gps to prevent molecules from entering the brain. OMA is considered a moderately hydrophilic compound due to its logP value of 1.88. This hypothesis can be further supported by the B/P ratio of OMA across these PDX models, ranging between 0.12 and 0.50 at 60 min (Table 2). These B/P ratio values can be considered low compared with other CNS-penetrant anti-cancer agents but comparable to temozolomide, the standard-of-care drug for glioblastoma, especially between 2 and 4 h post-drug administration (Figure 1D and Table 2) [41,42,43]. Notably, the tumoral concentrations of OMA observed in our three tested PDX models were significantly higher than those recently reported by Chen et al. [44] in a pediatric glioma model. Differences in the applied mouse model, tumor type and route of administration, i.p. versus s.c., may explain the differences in OMA pharmacokinetics.

A phase 1 study in glioblastoma patients and patients with brain metastasis conducted in the late 1980s by Savaraj et al., reported higher B/P ratios of tumor to plasma concentration of OMA, ranging from 0.5 to 1.8 [45], indicating differences in OMA metabolism between humans and mice. This variation can be attributed to different routes of administration and the elevated metabolic activity of esterases present in mouse plasma. Carboxylesterases-1d (Ces1d) is a murine isoform not present in humans [46], and is presumably responsible for the hydrolysis of ester bonds of OMA, ultimately leading to a rapid depletion of OMA in plasma. In the human brain, it is anticipated that CES expression would be low, explaining the slower CNS clearance of OMA and the increased B/P ratio over time. This hypothesis is further supported by the PK data in the GS832-PDX model, as illustrated in Table 2. However, using the B/P ratio in mice to assess the CNS penetration of OMA has inherent limitations, as it is influenced by the rate of plasma drug depletion and may not accurately represent local drug concentrations. Despite these constraints, the findings must be interpreted cautiously, as they could either underestimate or overestimate clinical relevance.

## 4. Conclusions

In conclusion, our study highlights both the challenges and opportunities in identifying drug candidates with favorable CNS penetration properties at the preclinical stage. The use of well-characterized orthotopic mouse PDX models for intracranial glioblastoma offers a robust platform for screening potential therapies, especially in light of the significant inter-tumoral heterogeneity observed across glioblastoma subtypes and their distinct BTB characteristics. These models provide critical insights that can guide the selection of promising candidates for clinical development in the treatment of glioblastoma. However, it remains important to recognize their limitations in predicting drug tumor exposure in humans, primarily due to factors such as murine metabolism. As we move forward, it becomes increasingly evident that human data on glioblastoma drug delivery and target modulation are essential. The growing interest in window-of-opportunity or phase 0 trials highlights the pressing need to address this issue before advancing novel agents to further clinical assessment [47]. These insights call for continued exploration and refinement of strategies for tackling this challenging disease.

## 5. Materials and Methods

### 5.1. Glioblastoma Stem-like Cell Cultures

Glioblastoma stem-like cell cultures (GSC) were derived from primary brain tumor samples received directly from the operating room. Tumor classification was performed by the local neuropathologist according to guidelines of the WHO 2007 and 2016 classification of primary brain tumors. The use of patient tissue for this study was approved by the local ethics committee and all patients signed informed consent forms according to the guidelines of the Institutional Review Board. Samples were processed as described previously [24]. The GBM8 cell line was kindly provided by dr. Hiroaki Wakimoto (Massachusetts General Hospital, Boston, MA, USA). All cell cultures were grown in Dulbecco’s modified Eagle’s medium (DMEM)–F12 with 1% penicillin/streptomycin, B27 (Invitrogen), human epidermal growth factor (EGF; 5 μg/mL), human basic fibroblast growth factor (FGF; 5 μg/mL) (both from Tebu-Bio) and heparin (5 mg/mL; Sigma-Aldrich, St. Louis, MI, USA).

### 5.2. Reagents

Omacetaxine mepesuccinate for in vitro and in vivo use was purchased from Sigma Aldrich. A stock solution of 20 mM dissolved in DMSO was prepared. Further dilutions for in vitro experiments were made in a serum-free medium for in vitro experiments and in a NaCl solution for in vivo experiments.

### 5.3. In Vitro Drug Screening

Glioma stem-like cell cultures were cultured under serum free conditions according to Verheul et al. [24]. Cells were seeded in 96-well plates (Greiner bio-one, Kremsmünster, Austria) coated with extracellular matrix (BD Bioscience, Becton Drive Franklin Lakes, NJ, USA) at 1000 cells/well. A stock solution of OMA was diluted in a serum-free cell culture medium to obtain a starting concentration of 8 μM in 0.1% DMSO. Serial dilutions of OMA and DMSO were added to the wells 24 h later for an incubation period of 5 days. Cell viability was measured by the CellTiter GLO 2.0 and dose–response curves and IC_50_ values were calculated using GraphPad Prism version 9. All experiments were performed in 3 biological replicates.

### 5.4. Statistical Analysis

GraphPad Prism software version 9.5 (GraphPad Software) was used for calculating IC_50_ values using nonlinear regression (curve-fit). GraphPad Prism was used to determine whether there was a statistically significant difference in OMA concentration and the B/P ratio between the L and R hemispheres of mice bearing GS832, GS607 and GBM8 tumors, using an unpaired *t*-test. For the GS32-PDX model, a two-way ANOVA was also employed to assess whether there were statistically significant differences in OMA concentration and the B/P ratio across three different time points within each hemisphere (Appendix A). *p*-values > 0.05 were considered not statistically significant.

### 5.5. Cell Lines and Transport Assays

In vitro transport studies were performed using the LLC pig kidney cell line (LLC-PK1) and sub-lines transduced with murine Abcb1a (LLC-Mdr1a) or human ABCB1 (LLC-MDR1). Similarly, in vitro transport studies were conducted in the parental Madin–Darby canine kidney (MDCK) type II cell line (MDCK-Parent) and murine Abcg2 (MDCK-Bcrp1) or human ABCG2 (MDCK-BCRP) transduced sublines. Concentration equilibrium transport assays (CETA) were performed as previously described [48]. Briefly, cells were grown on polycarbonate membrane filters (3.0 µm pores, 24 mm diameter; Costar Corning, NY, USA) for 4 days. Next, 2 mL of MEM medium containing 10% FCS and omacetaxine (100 nM) was added to both the apical and basal compartments. Then, [14C]-inulin (105 DPM/mL) was added basolateral to check the membrane integrity. Zosuquidar 1 μM was added to all wells containing MDCK cells to inhibit endogenous (canine) ABCB1. Plates were kept at 37 °C in 5% CO2 during the experiment, and samples of 100 μL were collected from both compartments at 5 min, 30 min, 1 h, 2 h and 4 h for analysis by liquid chromatography with tandem mass spectrometry detection (LC-MS/MS). The integrity of the membrane was checked by taking 20 μL samples from both apical and basolateral sides for radioactivity counting.

### 5.6. LC-MS/MS Analysis

An amount of 10 μL medium samples was vortex-mixed with 30 μL of acetonitrile: formic acid (100:1 *v*/*v*) containing 100 nM RO-3306 Internal Standard). After centrifugation, the supernatant was 5-fold diluted in water. OMA concentrations were measured using an LC-MS/MS system that consisted of an UltiMate 3000 LC System (Dionex, Sunnyvale, CA, USA) and an API 4000 mass spectrometer (Sciex, Framingham, MA, USA). Samples were analyzed using a ZORBAX Extend-C18 column (Agilent, Santa Clara, CA, USA), preceded by a Security-guard C18 pre-column (Phenomenex, Utrecht, The Netherlands). Elution was performed using a mixture of mobile phase A (0.1% formic acid in water (*v*/*v*)) and mobile phase B (methanol) in a 5 min gradient from 20% to 95% B, followed by 95% B that was maintained for 3 min and then re-equilibrated at 20% B. Multiple reaction monitoring parameters were 546.6/298.1 (OMA) and 352.1/186.0 (RO-3306).

### 5.7. Orthotopic Patient-Derived Xenograft Models

Non-obese diabetic severe combined immunodeficient (NOD/SCID) female mice 6–8 weeks of age were used for this study. All experiments were ethically approved by the local Animal Experiments Committee. Patient-derived glioblastoma cell lines were cultured under serum-free conditions prior to injection into the mice. Mice were stereotactically injected with GS607, GS832 (both 2.5 × 10^5^ in 5 μL) or GBM8 (1 × 10^5^ in 3 μL) in the right striatum 1.0 mm anteroposterior (AP), 2.0 mm medio lateral (ML) and −2.5 mm dorsal ventral (DV) of bregma suture under aseptic conditions [34]. Mice were monitored daily for signs of persistent abnormal behavior and/or breathing, weight loss >20% and development of neurological symptoms such as convulsions, seizures and/or hemiparesis. Mice were euthanized and the mouse brains harvested for histological and LC/MS/MS analysis.

### 5.8. Immunohistochemistry/Immunofluorescence

Mouse brains were fixated in 4% PFA solution and embedded in gelatin prior to sectioning. Coronal sections of 50 μm were stained for expression of Nestin (1:250, ab22035, Abcam, Cambridge, UK), GLUT-1 (1:1000, #PA1-46152, Thermo-Fischer, Waltham, MA, USA) and ZO-1 (1:100, #14-9776-82, Thermo-Fischer). DAPI was applied for staining DNA in the nucleus. Coronal sections of 10 μm specimen of snap-frozen mouse brains were stained for P-gp (1:200, 13978, Cell Signaling Technology, Danvers, MA, USA) and BCRP (1:400, ab24115, Abcam). The applied antibodies recognize both human and murine P-gp and BCRP. For histological analysis, coronal sections of 6 µm were fixated in acetone and stained with hematoxylin/eosin (H/E). Images were captured with the NanoZoomer (Hamamatsu) and analyzed with NDP viewer v.2 (Hamamatsu) whereas fluorescent images were captured using a Zeiss LSM700 confocal microscope (Carl Zeiss AG, Oberkochen, Germany) with a 40× oil immersion objective for our imaging. Z-stack imaging was performed, and the images presented were maximum Z-projections derived from these Z-stacks.

### 5.9. Magnetic Resonance Imaging (MRI)

All MRI imaging was performed on a preclinical 7T scanner (Discovery MR901, Agilent technologies, Santa Clara, California, USA) with a 300 mm bore diameter. MRI images were captured 70 days post-tumor implantation for GS607 and GS832-PDX, and 30 days for GBM8. Images were acquired with the 150 mm body coil for transmit and a four-channel phased-array surface receive coil (Rapid Biomedical, Würzburg, Germany). T1 weighted images were collected with a 2D fast spoiled gradient echo sequence, both before and after the administration of gadolinium contrast. T2-weighted imaging utilized a 2D fast spin echo sequence. For the GBM8 tumor mouse model specifically, a 3D approach was used. In addition, we obtained diffusion weighted images to study the viability of the brain tumor. Images were analyzed using the software Imalytics Preclinical 3.0 (Gremse IT, Aachen, Germany).

### 5.10. Pharmacokinetics of Omacetaxine

A single dose of 1.5 mg/kg of OMA was administered s.c. to glioblastoma-bearing NOD/SCID and naïve non-tumor-bearing mice. Mice received OMA prior to the timepoint drug delivery experiment, ensuring the presence of significant tumor masses. For GS607, this was at 78 days; for GS832 at 90 days; and for GBM8 at 46 days post-tumor implantation, based on previous survival experiments. Blood samples were collected at 15 min, 30 min, 60 min, 120 min and 240 min via the tail vein and/or submandibular vein and placed in EDTA tubes. Mice were euthanized at 60 min, 120 min or 240 min with cervical dislocation. Brains and livers were surgically collected and snap-frozen in liquid nitrogen (N2). To preserve the drug stability in plasma mouse blood, samples were centrifuged immediately upon collection. Plasma was separated by centrifugation at 3500 rpm at 4 °C for 15 min. Both plasma and tissue samples were stored at −80 °C until LC-MS/MS analysis.

## Figures and Tables

**Figure 1 pharmaceuticals-17-01687-f001:**
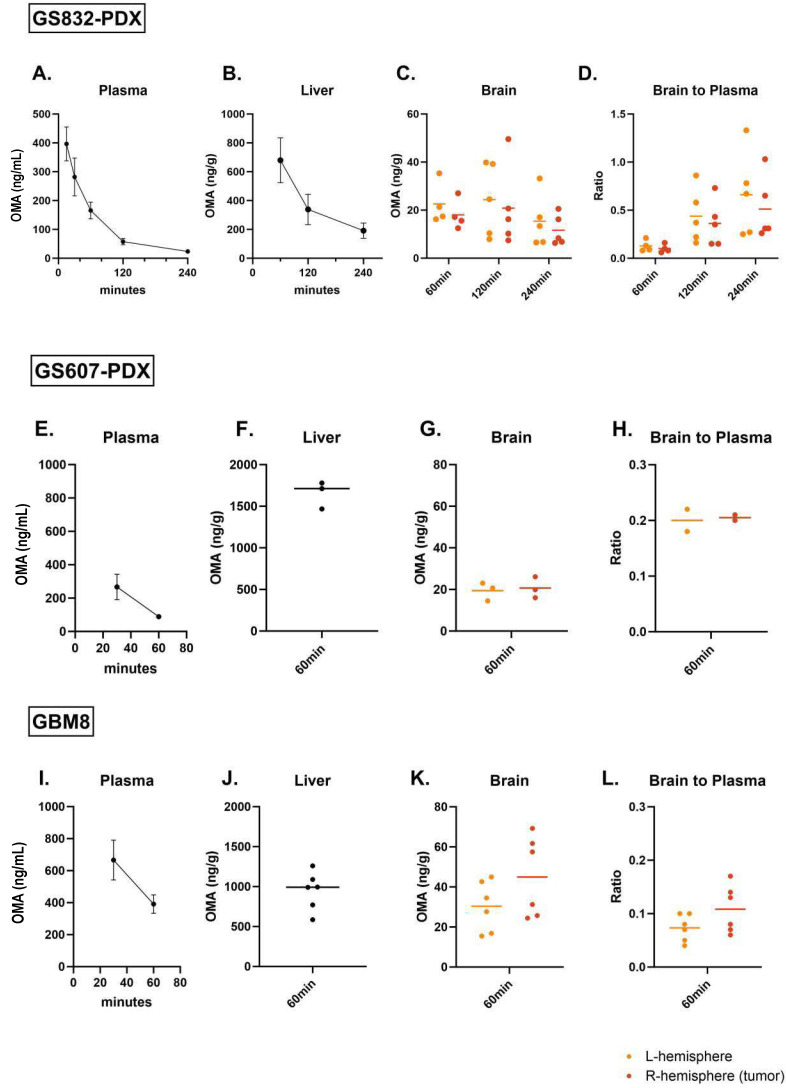
Pharmacokinetic analysis of OMA in vivo indicates accumulation in the brain tumor tissue. LC/MS/MS analysis determined the drug concentration after a single dose of 1.5 mg/kg of OMA in tumor-bearing mice. (**A**) Assessment of omacetaxine’s concentration in plasma (ng/mL), (**B**) liver (ng/g), (**C**) brain tumor tissue (ng/g) at 1, 2 and 4 h and (**D**) B/P ratio at 1, 2 and 4 h in GS832 PDX model (n = 5–10). (**E**) Assessment of drug concentration in plasma (ng/mL), (**F**) liver (ng/g) and (**G**) brain tumor tissue (ng/g) at 1 h and (**H**) B/P ratio at 1 h in GS607 PDX model (n = 4). Assessment of drug concentration in (**I**) plasma (ng/mL), (**J**) liver (ng/g) and (**K**) brain tumor tissue (ng/g) at 1 h and (**L**) B/P ratio at 1 h in GBM8 PDX model (n = 7).

**Figure 2 pharmaceuticals-17-01687-f002:**
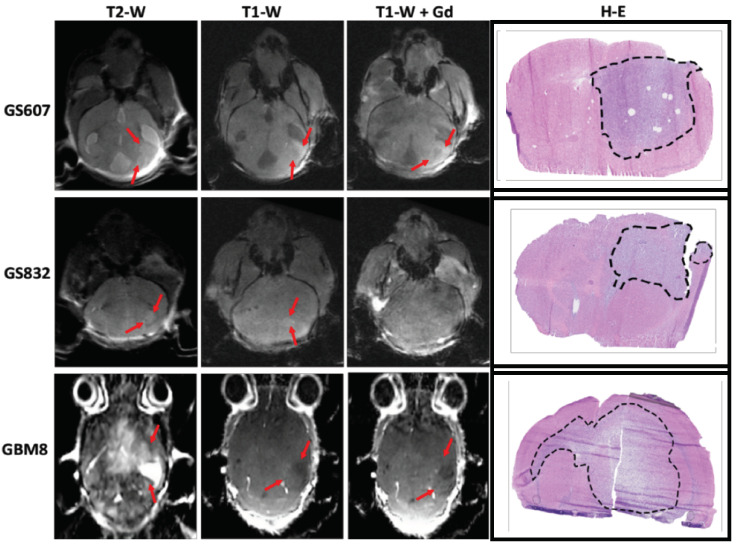
Assessment of the BBB permeability in three glioblastoma PDX models. The permeability of the BBB was evaluated by T2W, T1W pre- and post-gadolinium MRI imaging. Histological visualization of the tumor by H/E staining of mouse brain coronal slices. MRI images were obtained 70 days post-tumor implantation for GS607 and GS832-PDX, and 30 days post-implantation for GBM8. Black dashed lines indicate the tumor areas in the mouse brain slices, while red arrows denote the tumor regions in the MRI images. The number of mice used in this experiment was 3 per model.

**Figure 3 pharmaceuticals-17-01687-f003:**
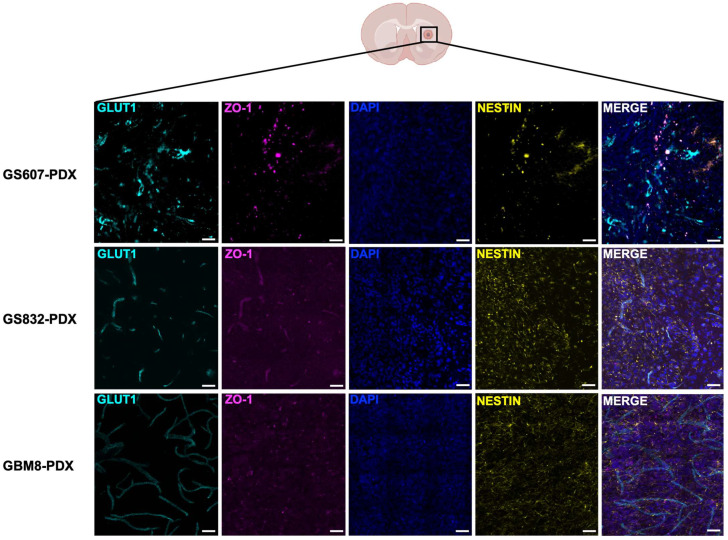
Characterization of the vascular integrity of the glioblastoma PDX models, GS607, GS832 and GBM8. The integrity of the vasculature was assessed by immunofluorescence staining for ZO-1 and GLUT-1, and GSC infiltration by Nestin staining. The number of mice used in this experiment varied between 3 and 6 per model. Scale bar 100 μm.

**Figure 4 pharmaceuticals-17-01687-f004:**
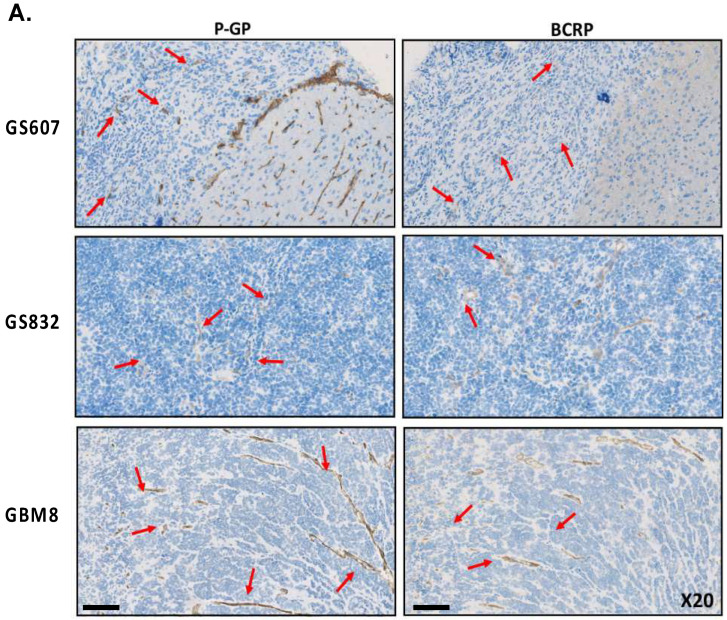
(**A**) Assessment of P-gp and BCRP transporter’s expression in the glioblastoma PDX models by IHC staining, magnification ×20 and scale bar 100 μm. Red arrows indicate the expression of the P-gp and BCRP transporters in the mouse brain tumor areas. The number of mice used in this experiment varied between 3 to 8 per model. (**B**) Assessment of omacetaxine’s in vitro affinity for P-gp and BCRP transporters by LC/MS-MS. The graphs depict the concentration of OMA as a percentage (*y*-axis) in the basal and apical compartments of LLC cells expressing murine Abcb1a and human ABCB1 and MDCK cells expressing murine Abcg2 and human ABCG2, over time in hours (*x*-axis). Zosuquidar was added to MDCK cells to inhibit endogenous (canine) P-gp. All experiments were performed in 3 biological replicates.

**Table 1 pharmaceuticals-17-01687-t001:** Plasma levels of single dose of 1.5 mg/kg of omacetaxine in NOD/SCID mice.

Timepoints (minutes)	[Plasma] ng/mL ± SD	Number of Samples
15	396.3 ± 58.7	n = 9
30	282.0 ± 65.4	n = 10
60	165.7 ± 28.7	n = 9
120	57.8 ± 10.8	n = 10
240	23.9 ± 2.86	n = 5

The values above represent the mean concentration of omacetaxine in the GS832-PDX model at 60, 120 and 240 min post-administration of a single dose of 1.5 mg/kg of omacetaxine.

**Table 2 pharmaceuticals-17-01687-t002:** Pharmacokinetic analysis of 1.5 mg/kg of omacetaxine in glioblastoma PDX models.

PDX-Model	Timepoints (minutes)	[Liver] ng/g ± SD	[Brain] † ng/g ± SD	[Plasma] ng/mL ± SD	[Brain]/[Plasma] Ratio ± SD
GS832	60	680.0 ± 155.9	18.0 ± 6.3	186.0 ± 21.1	0.10 ± 0.05
	120	338.2 ± 94.0	20.8 ± 16.9	55.6 ± 10.7	0.37 ± 0.24
	240	190.1 ± 47.8	11.6 ± 6.4	23.5 ± 0.7	0.50 ± 0.33
GS607	60	1653.4 ± 163.6	20.7 ± 5.05	88.4 ± 9.0 ^‡^	0.23 ± 0.01 ^‡^
GBM8	60	998.2 ± 254.3	45.0 ± 20.0	391.3 ± 57.5	0.11 ± 0.04
Naïve	60	715.9 ± 4.6	16.4 ± 2.9 *	224.5 ± 12.0	0.05 ± 0.01

The values above represent the mean concentration of OMA in the different glioblastoma mouse models at 60, 120 and 240 min post-administration of a single dose of 1.5 mg/kg of omacetaxine in GS832-PDX (n = 5) and at 1 h post-administration in GS607-PDX (n = 4), GBM8 (n = 6) and naïve mice (n = 2). † Right hemisphere (tumor bearing), * right hemisphere (non-tumor-bearing), ‡ (n = 2) blood samples at 60 min.

## Data Availability

The original contributions presented in this study are included in the article/Appendix A. Further inquiries can be directed to the corresponding author.

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
