# Peer review of "A Repurposed Drug Selection Pipeline to Identify CNS-Penetrant Drug Candidates for Glioblastoma"

_pharmaceuticals, 2024, doi:10.3390/ph17121687_

Round 1

Reviewer 1 Report

Comments and Suggestions for Authors

The manuscript entitled “A repurposed drug selection pipeline to identify CNS penetrant drug-candidates for glioblastoma” describes the findings of drug delivery parameters in three different orthotopic mouse PDX models using Omacetaxine mepesuccinate for its anti-glioma activity and CNS-MPO score. It has some experimental studies supporting the research’s idea but there are important points should be considered before accepting for publication as the following:

1 1-Abstract:

Some terms were expressed in an abbreviated form like BBB and some terms were expressed as both prolonged and abbreviated like Omacetaxine mepesuccinate (OMA). Unify the writing way through the whole manuscript by writing both prolonged and abbreviated forms first time, then write abbreviated after this. Change IC50 to IC50

22-Introduction and study’s rationale:

-The citation method doesn’t follow the journal’s format. Return to the author’s instruction guidelines and follow the correct way of citation.

-The rationale of this study needs some clarification about the following points:

-Justify using these three orthotopic patient-derived (PDX) models of glioblastoma.

-The criterion for selection of the FDA-approved Oncology Drug Set II library

 -The clinical importance of this study and how to employ it clinically.

33-Statistical analysis:

Write more details about the statistical analysis, like how many times the experiments were performed, how are the results expressed, and what test was used for comparing groups.

44-Conclusion:

There is no conclusion in the manuscript. Add a short paragraph concludes the main findings, clinical applications, and future perspective for it.  

55-References:

They also do not follow the journal’s format. Return to the author’s instruction guidelines and follow the correct way.

66-There are many typing mistakes in the manuscript like:

IC50: sometimes written as IC50 and sometimes written as IC50, correct it through whole manuscript, put a space between the number and the unit like: 43nM is changed to 43 nM, 396±59 to 396 ± 59.

Carefully review the whole manuscript and correct these mistakes.

Author Response

Reviewer 1:

We thank the reviewer for critical reading of our manuscript and providing insightful feedback to improve it.

  • Abstract: Some terms were expressed in an abbreviated form like BBB and some terms were expressed as both prolonged and abbreviated like Omacetaxine mepesuccinate (OMA). Unify the writing way through the whole manuscript by writing both prolonged and abbreviated forms first time. Then write abbreviated after this. Change IC50 to IC50

The use of abbreviations has been updated throughout the manuscript and we have adapted IC50 to IC50.

  • Introduction and study rationale:
  1. The citation method doesn't follow the journal's format Return to the author's instruction guidelines and follow the correct way of citation.

The rationale of this study needs some clarification about the following points:

  1. Justify using these three orthotopic patient-derived (PDX) models of glioblastoma
  2. The criterion for selection of the FDA-approved Oncology Drug Set II library
  3. The clinical importance of this stud and how to employ it clinically

  1. The citation style has been adjusted based on the MPDI guidelines.
  2. The three PDX models were selected based on their representation of GBM heterogeneity in terms of growth rates, growth patterns, BBB integrity and GBM subtypes (mesenchymal and (pro)neural). In the lines 89-93 we justified the reason for selecting and using these three PDX-models of glioblastoma.
  3. Over the past 10 years, our lab has generated a wealth of in vitro data using the FDA-approved Oncology Drug Set II Library which contains most of the commonly applied FDA-approved oncological drugs, and is generously provided by the NIH. This manuscript highlights the significance of our drug selection pipeline, which is designed to serve as a bridge for further in vivo validation of numerous promising candidates from the FDA-approved Oncology Drug Set II Library.
  4. In the lines 113-119 we highlighted the clinical importance of this study and how to employ it clinically.

  • Statistical analysis: Write more details about the statistical analysis, like how many times the experiments were performed. how are the results expressed, and what test was used for comparing groups?

Thank you for pointing this out. The statistical section (lines 410–416) has been updated, and a summary of the statistical tests is now provided as a supplementary file. Additionally, in lines 171–176, 204, 232–233, 260–261, 265–266, and 407, we have included details on the number of animals used in each experiment as well as the number of times each experiment was performed.

  • Conclusion: There is no conclusion in the manuscript. Add a short paragraph concluding the main findings, clinical applications and future perspective for it.

A conclusion section has been added in the manuscript, please see lines 366 -380.

  • References: They also do not follow the journal's format. Return to the author's instruction guidelines and follow the correct way.

The references have been updated according to MDPI’s style.

  • Typing mistakes: There are many typing mistakes in the manuscript like:

IC50: sometimes written as IC50 and sometimes written as IC50, correct it through whole manuscript. Put a space between the number and the unit like: 43M is changed to 43 nM

Carefully review the whole manuscript and correct these mistakes

The manuscript has been reviewed carefully and the typing mistakes have been corrected following your suggestion.  

Reviewer 2 Report

Comments and Suggestions for Authors

In the current manuscript by Ntafoulis et al, significant and substantial in vitro & in vivo data were presented. However unfortunately, I cannot capture the focus of the work presented well. 

In my understanding, the authors want to describe & propose a preclinical study plan to identify new or repurposed drug candidates for Glioma tumor therapy. Here are my concerns should this be the authors’ intention:

1) The authors chose three Glioma cell lines as PDX model, but it is relatively unclear the differentiation between these three models. I do not think these three PDX models are representative enough to cover the majority of Glioma tumors.

2) OMA is the only drug tested with further details in the proposed study plan. While I understand OMA shows the greatest score in the in silico prediction, it is hard to judge translatability of preclinical model when there is only one drug candidate tested, especially in the widely diverse Glioma tumor. 

3) Regarding the PK data in the three PDX model, I also could not understand the very wide differences from the plasma & liver sample at different timepoints. I could not understand how there is significant impact from the orthotopic injected tumor into the systemic distribution of the OMA.

I strongly suggest for the authors to clarify on these points before I can recommend for the manuscript to be accepted.  

Author Response

Reviewer 2:

We thank the reviewer for critical reading of our manuscript and providing insightful feedback to improve it.

1) The authors chose three Glioma cell lines as PDX model, but it is relatively unclear the differentiation between these three models. I do not think these three PDX models are representative enough to cover the majority of Glioma tumors.

We understand your concern about the representativeness of the three PDX models used in this study. While we agree that a larger number of models might better capture the intertumoral heterogeneity of glioblastoma, the primary focus of this research was to validate our drug selection platform in vivo and to investigate the pharmacokinetic (PK) profile of omacetaxine mepesuccinate as a candidate capable of crossing the BBB and penetrating brain tumor tissue.

To address differentiation, the three PDX models were selected based on distinct characteristics relevant to glioblastoma, such as their invasive properties, variability in BBB integrity, and ABC transporter expression. These features were deliberately chosen to reflect key biological barriers to CNS drug delivery, which are critical to our study's objectives. We have clarified in the text (lines 92–93) that all three models mimic the invasive characteristics of glioma cells, and that these models have differences in growth rate and Verhaak subtype. Additionally, the differences among these three models are highlighted in their variations in tumor vascularization (lines 206–228), BBB integrity (lines 192–198), and ABC transporter expression (lines 236–256) and glioblastoma subtypes (lines 184-185, 187 and 189-190). These characteristics are summarized for each model in Supplemental Table 2.

For these reasons, we believe that the models used are sufficient to address the specific research questions of this manuscript, while acknowledging that broader representation of glioma heterogeneity will be an essential focus of future therapeutic efficacy studies.

2) OMA is the only drug tested with further details in the proposed study plan. While I understand OMA shows the greatest score in the in-silico prediction, it is hard to judge translatability of preclinical model when there is only one drug candidate tested, especially in the widely diverse Glioma tumor.

We agree that addressing glioma treatment requires multiple candidates, given the substantial intra- and intertumoral heterogeneity of this disease. As noted in our previous response, the main objective of this manuscript is to validate our preclinical drug selection platform in an in vivo setting, focusing on its potential to identify and prioritize drug candidates for glioblastoma treatment.

In lines 354-361, we also discuss the limitations of preclinical models, particularly regarding species-specific metabolic differences that may influence the pharmacokinetics and hence efficacy of drug candidates. Therefore, we also emphasize the importance of phase 0/window-of-opportunity clinical studies to assess drug delivery and target modulation in validating preclinical findings and advancing promising compounds for clinical evaluation (lines 367-380).

3) Regarding the PK data in the three PDX model, I also could not understand the very wide differences from the plasma & liver sample at different timepoints. I could not understand how there is significant impact from the orthotopic injected tumor into the systemic distribution of the OMA.

As illustrated in Figure 1 of the manuscript, OMA shows a tendency to accumulate more in the liver than in plasma, which can be attributed to several key pharmacokinetic and pharmacodynamic factors. Primarily, OMA is metabolized in the body through non-enzymatic hydrolysis, although certain clearance mechanisms may involve conjugation processes within liver cells, promoting retention over plasma circulation. Additionally, the liver, being highly vascularized, has a unique capability to store and process compounds, especially those administered systemically. Upon administration, OMA rapidly reaches the liver, where it accumulates due to the organ’s rich blood supply and tissue-specific binding properties—possibly explaining the higher concentrations observed in all three tested PDX models compared to plasma.

Further, as a hydrophobic compound, OMA may be more readily absorbed and retained in liver tissue, as liver cell membranes are well-suited to absorb and store hydrophobic molecules. Consequently, this results in higher drug concentrations in the liver relative to the bloodstream. Our findings also indicate that OMA is rapidly cleared from plasma due to high esterase activity (lines 354–355), which leads to a faster decline in plasma concentrations than in liver concentrations. Since omacetaxine metabolism does not depend on enzyme systems like cytochrome P450, it may preferentially accumulate in tissues like the liver, where hydrolysis is effective, contributing to the observed difference in concentration levels between plasma and liver tissue.

Addressing the second part of your comment, Table 1 of this manuscript demonstrates that the systemic distribution of omacetaxine in orthotopic PDX models differs from that in naïve mice (those without brain tumors). This variance may be due to glioblastoma’s effects on systemic blood flow and organ function, particularly if the tumor increases intracranial pressure or induces stress responses. These conditions could alter circulation and impact the distribution and metabolism of omacetaxine in other tissues. We have included this potential explanation in the discussion (lines 315-319) to provide further context for these observations.

Round 2

Reviewer 1 Report

Comments and Suggestions for Authors

The authors addressed all the review comments, so I approve the manuscript for publication.

Reviewer 2 Report

Comments and Suggestions for Authors

The authors have sufficiently answered my queries and amended the manuscript accordingly.

I have no further queries.